# Effect of Serial Systemic and Intratumoral Injections of Oncolytic ZIKV^BR^ in Mice Bearing Embryonal CNS Tumors

**DOI:** 10.3390/v13102103

**Published:** 2021-10-19

**Authors:** Raiane Oliveira Ferreira, Isabela Granha, Rodolfo Sanches Ferreira, Heloisa de Siqueira Bueno, Oswaldo Keith Okamoto, Carolini Kaid, Mayana Zatz

**Affiliations:** 1Centro de Estudos do Genoma Humano e Células-Tronco, Departamento de Genética e Biologia Evolutiva, Instituto de Biociências, Universidade de São Paulo, Cidade Universitária, São Paulo 05508-090, SP, Brazil; raianeferreira@ib.usp.br (R.O.F.); isabelagranha@usp.br (I.G.); rsanches@ib.usp.br (R.S.F.); siqueirabueno@gmail.com (H.d.S.B.); keith.okamoto@usp.br (O.K.O.); 2Hemotherapy and Cellular Therapy Department, Hospital Israelita Albert Einstein, São Paulo 05652-900, SP, Brazil

**Keywords:** virotherapy, ZIKA virus, CNS tumors, murine pre-clinical study, immune cytokine profile

## Abstract

The Zika virus (ZIKV) has shown a promising oncolytic effect against embryonal CNS tumors. However, studies on the effect of different administration routes and the ideal viral load in preclinical models are highly relevant aiming for treatment safety and efficiency. Here, we investigated the effect and effectiveness of different routes of administration, and the number of ZIKV^BR^ injections on tumor tropism, destruction, and side effects. Furthermore, we designed an early-stage human brain organoid co-cultured with embryonal CNS tumors to analyze the ZIKV^BR^ oncolytic effect. We showed that in the mice bearing subcutaneous tumors, the ZIKV^BR^ systemically presented a tropism to the brain. When the tumor was located in the mice’s brain, serial systemic injections presented efficient tumor destruction, with no neurological or other organ injury and increased mice survival. In the human cerebral organoid model co-cultured with embryonal CNS tumor cells, ZIKV^BR^ impaired tumor progression. The gene expression of cytokines and chemokines in both models suggested an enhancement of immune cells recruitment and tumor inflammation after the treatment. These results open new perspectives for virotherapy using the ZIKV^BR^ systemic administration route and multiple doses of low virus load for safe and effective treatment of embryonal CNS tumors, an orphan disease that urges new effective therapies.

## 1. Introduction

The Zika virus (ZIKV) has been studied as a potential treatment against central nervous system (CNS) tumors [1] and several groups, including ours, have demonstrated its unique selective infection of neural stem-like cells. Therefore, it can also be a promising oncolytic therapy against CNS tumors closely resembling neural progenitor cells (NPCs), molecular fingerprints [2]. Our group demonstrated for the first time that the Brazilian ZIKV (ZIKV^BR^) can be an efficient oncolytic agent to treat aggressive forms of embryonal CNS tumors, including medulloblastoma and atypical teratoid rhabdoid tumors (ATRTs) [3]. These pediatric CNS tumors are enriched in stem-like cancer cells and are very difficult to treat since available therapies, including the new immunotherapies, have low efficiency and severe adverse effects, significantly affecting the quality of life of the few survivors. 

These recent studies indicate that ZIKV oncolytic therapy can be considered a promising treatment for these orphan diseases since the virus destroys tumor cells. Furthermore, it activates the immune responses during oncolytic viral infections, increasing immune cells infiltration and tumor microenvironment inflammation [4,5]. Before translating this potent virotherapy to clinical application, more investigation regarding the ZIKV^BR^ influence in normal and tumor environments is required to improve its oncolytic potential as well as the treatment safety. 

Here, based on our previous data of the ZIKV^BR^ oncolytic effect in mice bearing embryonal CNS tumors, we investigated the efficiency, possible neurological injury, and mice survival using different virus delivery routes as well as the effect of multiple doses. We show that serial systemic injections of ZIKV^BR^ are safe as it does not influence mice survival and does not cause high viremia and virulence. The serial systemic treatment in USP7ATRT tumor-bearing mice increased their survival rate without injuring healthy cells and improved the clinical signs. The in vitro human cerebral organoid model with embryonal CNS tumor cells, aiming to recapitulate human pathologies and microenvironment tumor complexity, confirmed that ZIKV^BR^ impaired tumor progression. Moreover, cytokines and chemokines analysis showed an increased expression of macrophage migration inhibitory factor (MIF) and Gpi1, a potent cytokine that plays a role as a metastasis inducer increasing the epithelial-mesenchymal transition (EMT) and tumor necrosis factor ligand superfamily member 13b (Tnfsf13b). 

## 2. Materials and Methods

### 2.1. Human and Animal Samples

The study followed the International Ethical Guideline for Biomedical Research (CIOMS/OMS, 1985) and was approved by the Institutional Animal Experimentation Ethics Committee (CEUA-USP nº 290/2017). Before euthanasia, the following clinical symptoms were observed in animals to minimize suffering: 30% weight loss and/or ataxia and/or visible tumor and/or freezing. The human induced pluripotent stem (hiPS) cell line used in this experiment was generated by reprogramming skin fibroblasts from a 26-year-old healthy man. The F9048 cell line presents pluripotent cell colony morphology and positive staining for markers OCT3/4 and SSEA-4, as previously established and characterized by our group [6]. 

### 2.2. Cell Lines and Culture

The F9048 hiPS cell line was cultured in Matrigel (Corning, Corning, NY, USA) coated 60 mm Petri dishes, in Essential 8 medium (Gibco, Thermo Fischer Scientific, Waltham, MA, USA), supplemented with 100 µg/mL Normocin (InvivoGen, San Diego, CA, USA), at 37 °C and 5% CO_2_ atmosphere. StemPro Accutase (Thermo Fisher Scientific, Waltham, MA, USA) was used for cell detachment. The commercial cell line Vero was cultivated according to ATCC recommendations. The embryonal CNS tumor cell lines USP7ATRT (atypical teratoid/rhabdoid tumor, in-house established [3], USP13MED (medulloblastoma, in-house established) [7] were cultured using Dulbecco’s modified Eagle’s medium (DMEM) supplemented with 10% fetal bovine serum (FBS) (Gibco, Thermo Fischer Scientific, Waltham, MA, USA), 100 U/mL Penicillin, 100 µg/mL Streptomycin and 250 ng/mL Fungizone^®^ (Gibco) at 37 °C at 5% CO_2_ atmosphere, and tested for *Mycoplasma* contamination by PCR (Sigma-Aldrich, Merck, Darmstadt, Germany), before use in the described experiments.

### 2.3. 3D Brain Model and Co-Culture Assay

Brain organoid development from hiPS cells was performed according to previously described protocols [8] using a STEMdiff^TM^ Cerebral Organoid Kit (StemCell Technologies, Vancouver, BC, Canada) and following the manufacturer’s instructions. Briefly, F9048 hiPS cell colonies were gently dissociated using Accutase (Thermo Fisher Scientific, Waltham, MA, USA) and counted using Trypan Blue in a Countess II^TM^ automated cell counter (Thermo Fischer Scientific, Waltham, MA, USA). A total of 9000 hiPS cells per well were plated in a 96-well round bottom ultra-low attachment plate in embryoid body (EB) seeding medium and cultured for 5 days, adding EB formation medium at days 2 and 4. Formed EBs were then transferred to 24-well ultra-low attachment plates in induction medium. After 3 days of neuroectodermal differentiation, spheres were embedded in Matrigel (Corning) and maintained in expansion medium for neuroepithelial bud outgrowth. On protocol day 10, the medium was changed to the maturation medium, and the plates were cultured on an orbital shaker at 65 RPM. The complete medium change was performed every 4 days. For all steps, plates were maintained at 37 °C and 5% CO_2_ in a humidified incubator. 

The co-culture of embryonal CNS tumor cells with immature (26-day-old) brain organoids was performed as follows. The 10^3^ USP7ATRT or USP13MED stably expressing GFP were added per organoid in 96-well round bottom ultra-low attachment plates, in the maturation medium. The medium exchange was performed every 2 days. Fluorescent microscopy images were obtained at days 0, 7, 14, and 21 post-co-culture, and fluorescence intensity was accessed using the ZEN software (2.1 blue edition, Carl Zeiss, Oberkochen, Germany) at day 21. The isolated 26-day-old brain organoids were used as control groups. 

The ZIKV^BR^ infection of experimental groups was performed on days 7, 14, and 21 post-co-culture. Co-cultured organoids were infected with one or three doses of 2000 ZIKV^BR^ plaque-forming units (PFUs) or pure maturation medium (mock group) for 1 h at room temperature. The virus containing media was removed and a fresh maturation medium was added. The supernatant medium was collected at days 7, 14, and 21 post-infection (DPI) for PFU and PCR assays. 

### 2.4. In Vivo Injection Assay

The in vivo assays were performed in Balb/C nude mice. The ZIKV^BR^ (2 × 10^3^ PFU/ZIKV^BR^) was injected systemically, intracranially, or intratumorally. The systemic injections were via the intraperitoneal pathway, while the intracranial injections were at the right lateral ventricle. The intratumoral injections were at the tumor located in the animal’s flank. The 10^6^ tumor cells, USP13MED or USP7ATRT, were injected at the right posterior animal’s flank or intracranially at the right lateral ventricle as described next. A total of 99 animals were included in the present study with about 8 animals per group. 

### 2.5. Orthotopic Metastatic Xenograft Animal Model

Tumor injections and infections with ZIKV^BR^ were performed in animal models as previously described [9]. The 10^6^ cells of the USP13MED or USP7ATRT lineage were injected intracranially into the right ventricle of the animals. After the establishment of the tumor, according to the growth kinetics of each cell line (1 week for USP7ATRT, 2 weeks for USP13MED), the animals received, peritoneally or intracranially, one or three doses of 2 × 10^3^ particles of ZIKV^BR^. Doses were given 1 week apart. Control groups were established with a systemic (via intraperitoneal)/intracranial PBS application. The animals were weighed every 2 days until weight loss was >30% and/or a visible tumor and/or ataxia were observed, then they were euthanized. 

### 2.6. ZIKV Strain and Viral Detection 

The ZIKV strain BR (GenBank accession number ID: KU497555) donated by Dr. Pedro Vasconcelos (Evandro Chagas Institute, Belém, Brazil) was propagated in Vero cells and the virus stocks were titrated by PFU assays.

The organoids culture supernatant from in vitro assays and blood, brain, spleen, ovary, or testis from in vivo assays were used for the viral detection analysis (PFU and PCR). Organs were weighed and macerated using the Precellys Evolution homogenizer equipment (Bertin Instruments, Montigny-le-Bretonneux, France). The blood serum was prepared after coagulation at 37 °C/30 min and 3000 RPM/10 min centrifugation. For PFUs, 10^5^ Vero cells were seeded in 24-well plates, 24 h before assay. The samples were serially diluted in the DMEM medium from 10^−1^ to 10^−12^, and applied in duplicates of 100 µL. Vero cells were infected for 1 h at 37 °C. After that period, the wells were covered with DMEM, 1% of the CMC medium (Sigma-Aldrich, Merck, Darmstadt, Germany) and incubated at 37 °C. After 4 days, the plates were washed with PBS and stained with crystal violet (Synth, Labsynth, Diadema, Brazil). PFUs were visually established in the most appropriate viral dilution and expressed as PFU/mL. 

The viral RNA was extracted from samples using the Viral RNA Mini Kit (Qiagen, Germantown, MD, USA) following the manufacturer’s recommendations. Viral RNA was quantified using One Step Taqman RT-qPCR (Thermo Fisher Scientific, Waltham, MA, USA) on a 7500 Real-Time PCR system (Applied Biosystems, Thermo Fischer Scientific, Waltham, MA, USA) with primers (ZIKV1086 Foward CCGCTGCCCAACACAAG; ZIKV1162 Reverse CCACTAACGTTCTTTTGCAGACAT; ZIKV007 probe 1 5′FAM AGCCTACCTTGACAAGCAGTCAGACACTCAA3′ BHQ1) designed by (Exxtend, Paulínia, Brazil). 

### 2.7. Histopathology and Immunofluorescence 

The brains and organoids for immunohistochemistry were embedded in 30% sucrose (Synth, Labsynth, Diadema, Brazil), in 0.1 M PBS, and cryosectioned at 10 µm with a cryostat (Leica, Wetzlar, Germany). At least three brains or organoids from each group, and three sections per brain or organoids were used for all immunohistochemistry analyses. Sections were washed in 0.1 M PBS, and incubated in blocking solution (90 min, 5% BSA (Sigma-Aldrich, Merck, Darmstadt, Germany) and 0.1% Triton-X100 (Sigma-Aldrich, Merck, Darmstadt, Germany) in 0.1 M PBS, and then primary antibodies (NS2B, GeneTex Cat. GTX133308, 1:500) at 4 °C overnight. Sections were washed and incubated with a fluorescent-conjugated secondary antibody (1 h, Alexa Fluor 546 anti-rabbit, Invitrogen, cat, 11010, at 1:500 dilution) and stained with 4′-6-diamino-2-phenylindole (DAPI, Invitrogen, 1:1000 dilution in 0.1 M PBS) nuclear counterstain. The cell death assay was performed with the ApopTag Apoptosis Kit (Sigma, Merck, Darmstadt, Germany) following the manufacturer’s recommendations.

### 2.8. Real-Time Quantitative PCR Array 

To perform the PCR array assays, the total RNA was extracted from the organoids and mouse brains using the RNeasy Mini Kit (Qiagen, Germantown, MD, USA). The brains were macerated using the Precellys Evolution homogenizer equipment (Bertin Instruments, Montigny-le-Bretonneux, France). The cDNA synthesis was performed using the RT^2^ First Strand Kit (Qiagen, Germantown, MD, USA) and the cytokines and chemokines from in vitro (GeneGlobe ID—PAHS-150Z) or in vivo (GeneGlobe ID—PAMM-150Z) samples were quantified by the RT^2^ Profiler^TM^ PCR Array Kit (Qiagen, Germantown, MD, USA) following the manufacturer’s instructions.

### 2.9. Image and Statistical Analysis 

Images were taken using Evos XL or a confocal laser-scanning microscope, Zeiss LSM 800.

The data were analyzed by a one-way and two-way ANOVA followed by a Bonferroni post hoc test. The *t*-test with a two-tailed unpaired test was used for a pairwise comparison. The clinical and systemic viral infection parameters were analyzed by the Fisher exact test. The GraphPad Prism software was used to perform all statistical analyses (version 6.0 GraphPad Software Inc.). The quantification of data is represented as mean ± SEM, and the *p*-value threshold was as follows: *, ≤0.05; **, ≤0.01; ***, ≤0.001; and ****, ≤0.0001.

## 3. Results

### 3.1. Evaluation of Viral Safety in Balb/c Nudemice after Serial Systemic and Intracerebroventricular ZIKV^BR^ Infections

First, aiming to analyze the safety of ZIKV^BR^ of internalization methods, Balb/c nude tumor-free mice were infected three times with a low viral load through systemic injections (2 × 10^3^ PFU/ZIKV^BR^) with an interval of 7 days between each dose (Figure 1A). No significant weight loss and impaired clinical signs were observed in ZIKV^BR^ infected animals compared with the control group (Figure 1B). In addition, ZIKV^BR^ non-structural protein NS2B was not detected in the brain tissue histological analyses (Appendix A), and viral RNA was also not detected by the qPCR. When testing the serial ZIKV^BR^ intracerebroventricular (ICV) injections in Balb/c nude mice infected with the same viral load, we observed a significant weight loss in infected animals compared with the control group that received the buffer solution (Appendix A). Infected animals survived up to 4 weeks after ICV viral administration, while the control group remained alive and without clinical changes (Appendix A). The viral RNA was detected in the blood, brain, and spleen by qPCR in infected animals, in which it was not detected in the control group (Appendix A). These results indicate that serial ZIKV^BR^ ICV injections induce high virulence and viremia in these mice, resulting in death a few days after the last infection (17–28 days).

### 3.2. Evaluation of Viral Tropism in Mice Bearing Subcutaneous Embryonal CNS Tumor after Systemic ZIKV^BR^ Infections

The ZIKV tropism to CNS is well-established in the literature [10]. However, there are no data assessing the viral preference towards the CNS with or without the tumor. So, we analyzed whether the virus tropism would be directed to the CNS or to the NPC-like tumor cells (USP7ATRT) if the tumor mass was located subcutaneously. For this, Balb/c nude mice received 10^7^ cells of USP7ATRT at the right flank (Figure 1C). After the tumor reached 1 mm^3^, the animals were treated with systemic or intratumoral injections and with ZIKV^BR^ through a single or multiple doses (3 doses) with the same viral charge used before (2 × 10^3^ PFU/ZIKV^BR^). Although no significant variation was observed, the tumor weight average of the ZIKV^BR^ intratumoral group was smaller when compared with its respective control group (Figure 1E). The intratumoral ZIKA^BR^ treated group had significantly slower tumor growth, smaller tumor volume accompanied by a longer period of doubling time, than systemic treated or mock groups (Figure 1D,F–I). Additionally, the intratumoral injection induced an intense necrotic hole in the tumor mass (Figure 1J–L). Besides that, no tumor remission was observed in both systemically treated groups (Figure 1D,F,H), suggesting that the ZIKV tropism is directed to CNS. Curiously, the tumor volume was smaller and had a longer doubling time in the group treated with a single intratumoral viral dose (Figure 1F) (doubling time in days: mock = 6.144; systemic injection = 8.755; intratumoral injection = 13.95) compared with the group that received three doses (Figure 1D) (doubling time in days: mock = 6.849; systemic injection = 6.680; intratumoral injection = 7.863).

### 3.3. ZIKV^BR^ Oncolytic Effect in Intracranial Tumor-Bearing Mice Varying ZIKV^BR^ Internalization

As the viral tropism for the mice CNS and systemic injection safety was confirmed (Figure 1), we investigated whether the systemic ZIKV^BR^ injections would lead to a viral invasion in the intracranial tumors. To address this question, we injected 10^6^ cells from USP13MED or USP7ATRT at the right lateral ventricle of Balb/c nude mice (Figure 2A). After the tumor establishment, three doses of ZIKV^BR^ (2 × 10^3^ PFU/ZIKV^BR^) (treatment) or PBS (control) were systemically injected with a 7 day interval between doses. It was not possible to observe weight loss or significantly different survival rates between the treated and control USP13MED-bearing-group (Figure 2B,C). The qPCR did not detect the viral RNA in the brain of infected animals (Figure 2D). 

On the other hand, mice bearing intracranial USP7ATRT tumors showed weight loss with a significantly higher survival rate in treated as compared with untreated controls (Figure 2E,F). The ZIKV^BR^ RNA was detected through qPCR in the blood, spleen, and brain of the infected animals (Figure 2G). Most importantly, we observed neither the virus (NS2B, red) nor cell death (TUNEL, green) at the healthy brain cortex of the treated animals through immunofluorescence (Figure 3A–D). Yet, the immunofluorescence from the edge of the USP7ATRT tumor showed positive cells for ZIKV^BR^ and increased cell death in treated animals compared with non-treated groups (mock) (Figure 3E–L). Besides that, it was possible to observe that the ZIKV^BR^ tag (NS2B, red) overlapped with the cell death tag (TUNEL, green) in the tumor treated group (Figure 3L–L’’’, yellow dots) indicating that the cell apoptosis had co-located with the virus presence. In addition, the animals in both experimental groups were euthanized following the criteria for weight loss (<70% of initial weight) and clinical symptoms, such as ataxia. Most ZIKV^BR^-treated mice bearing USP7ATRT tumors were euthanized due to weight loss, still showing better exploratory behavior and clinical signs compared with untreated mice bearing USP7ATRT tumors (Video 1).

Since we observed no survival improvement in mice bearing intracranial USP13MED cells after systemic injection, we questioned if intratumoral ZIKV^BR^ injections can show similar effectiveness as shown in the subcutaneous tumor model (Figure 1B). So, we decided to investigate the effect of intracerebroventricular (ICV) ZIKV treatment, using a virus internalization method that mimics an intratumoral injection for embryonal CNS tumors.

For that, the orthotopic xenograft animal model in Balb/c nude mice previously described by our group [7] with USP13MED and USP7ATRT cell lines was used to evaluate the ZIKV^BR^ oncolytic effect with intracerebroventricular (ICV) multiple injections. The experiment was performed with three doses of ZIKV^BR^ (2 × 10^3^ PFU/ZIKV^BR^)-injected in the right lateral ventricle with an interval of 7 days between each injection. The injections started after the time required for tumor establishment, 7 days and 14 days for USP7ATRT and USP13MED tumor cells, respectively. Tumor-bearing mice treated with ZIKV^BR^ intracranial showed significant weight loss and a low survival rate for both tumor cell lines (Appendix A–E). The viral RNA was detected in the blood, spleen, and brain of the infected animals (Appendix A). ZIKV^BR^ non-structural protein NS2B was detected in the cerebral cortex with intensive cell death inside the USP7ATRT tumor (Appendix A–H). In addition, abnormal brain parenchyma and pericellular edema were observed in USP7ATRT- and USP13MED-injected brains (Figure 3M´,O´), probably due to the high viremia still present in the region.

However, in the histological analysis no tumors were found in USP7ATRT and USP13MED tumor-bearing mice with ZIKV^BR^ ICV treated brains (Figure 3M,O), in comparison with the control group that presented large tumors (Figure 3N,P). Therefore, although ZIKV^BR^ ICV serial treatment confirmed the oncolytic effects for embryonal CNS tumor-bearing mice, serial doses resulted in high virulence and viremia, causing animal death. However, while Balb/C nude mice represent the possibility to study the oncolytic effect of human tumors in a murine model, its unique sensibility to viral infection due to thymus absence does not allow to translate safety findings to immunocompetent patients.

### 3.4. ZIKV^BR^ Infection Impairs Tumor Cells Spread in Brain Organoids In Vitro

It is known that animal models do not always recapitulate human pathologies, and the microenvironment tumor complexity requires different research approaches. Aiming to investigate the ZIKV^BR^ oncolytic selective infection in vitro, we developed early-stage (26 days) human cerebral organoids with high amounts of SOX2 positive cells (Appendix A). The brain organoids were co-cultured with 10^3^ green fluorescent protein (GFP) positive USP7ATRT or USP13MED cells and infected with 2000 PFU of ZIKV^BR^ 7 days after tumor cell addition (Figure 4A). GFP^+^ tumor cells rapidly attached and started to spread in brain organoids after 7 days (0 DPI) of co-culture (Figure 4B). Cancer cells appear to have clustered in tumor-like structures within the organoids. Interestingly, treatment of co-cultured models with one or three ZIKV^BR^ doses caused a pronounced reduction in GFP^+^ cell mass over time, for both USP13MED and USP7ATRT (Figure 4B,C) cell lines. Conversely, in the mock group, GFP^+^ tumor cells had taken over most of the brain organoids’ area by day 21 (Figure 4B). GFP intensity quantification of 21 DPI images reinforced the tendency of fluorescence reduction in ZIKV treated groups (Figure 4C). Treatment of USP13MED subgroups with one (*p* < 0.01) or three (*p* < 0.05) ZIKV^BR^ doses significantly reduced GFP intensity, when compared with equivalent mock groups (Figure 4B,C). Most importantly, viral infection confirmed by ZIKV^BR^ NS2B immunolabeling and positive TUNEL staining was found in tumor cells SOX2^−^, suggesting virus infection preference when compared with the stem-like normal cell, SOX2^+^ (Figure 4E–L).

To better demonstrate the ZIKV infection preference to tumor cells, we quantified the nucleus area of the GFP^+^ tumor cells, ZIKV positive (ZIKV^+^) area, and ZIKV negative (ZIKV^−^) area (Appendix A–G). Nuclear size has been used by cytopathologists as an important parameter for cancer prognosis. The most common changes in the tumor cells, when compared with normal cells, are related to the nucleus size [11]. We observed that the average size and the nuclei morphology present in the ZIKV^+^ area is similar to the GFP^+^ tumor cells (65.17 µm^2^ and 86.07 µm^2^, respectively). The organoid area without ZIKV staining had smaller nuclei with an average of 29.41 µm^2^ indicating non-tumor cells (Appendix A). In Appendix A we detailed better the ZIKV^+^ area with increased nucleus cells and ZIKV^−^ area with small nucleus cells, further suggesting ZIKV infection preference to tumor cells (increased nuclei), sparing the normal cells (smaller nuclei). Thus, in vitro infection of co-cultured models impaired tumor progression in all settings by infecting tumor cells indicating an intensive oncolytic effect of ZIKV^BR^ in CNS embryonal tumors in vitro. However, more investigation is required to confirm the virus selectivity.

Similar to CNS tumor cells, brain organoids progenitor cells are highly susceptible to ZIKV^BR^ infection [12,13] and upon infection of human cells, ZIKV^BR^ usually peaks infectious viral titers quite rapidly and starts to decay [14]. Our PFU analysis of early-stage organoids supernatants, co-cultured with and without tumor cells, revealed the same peak infection with a viral titer up to 10^10^ PFUs (Figure 4D). The reduction tendency of infectious viral particles over time was observed in all subgroups, except in the Organoid + USP7ATRT three-dose subgroup. Interestingly, the multiple doses showed no viral titer difference over time when compared with one-dose group. The additional 2000 PFU doses seem to present no significant effect since the culture supernatant viral titer is already extremely high (Figure 4D).

### 3.5. ZIKV^BR^ Infection in Tumor Cells Changes the Cytokines and Chemokines Expression In Vitro and In Vivo

Aiming to investigate the gene expression of cytokines and chemokines after ZIKV^BR^ administration in orthotopic metastatic xenograft animal model and in vitro 3D brain models, we used the RT-PCR array kit (Qiagen) containing diverse chemokines, interleukins, interferons, growth factors, TNF receptor superfamily members and anti-inflammatory cytokines. Surprisingly, the chemokine and cytokine expression profile of mice bearing an embryonal CNS tumor clustered with tumor-free mice with or without systemic ZIKV^BR^ serial administration (Figure 5A). The mice bearing a USP7ATRT tumor and treated with ZIKV^BR^ presented a cytokine and chemokine expression profile different from other groups. Expression of macrophage migration inhibitory factor (MIF) and Gpi1, a potent cytokine that plays a role as a metastasis inducer increasing the epithelial-mesenchymal transition (EMT) [15] were upregulated after tumor cell injection and downregulated to almost zero, after ZIKV^BR^ treatment (Figure 5B). The same downregulation was observed for Vegfa, a cytokine involved in the EMT, important for the tumor angiogenesis and metastatic processes [16]. When considering the opposite signaling, we observed an upregulation of Hc cytokine, tumor necrosis factor ligand superfamily member 13b (Tnfsf13b), and the anti-inflammatory cytokine IL-6 only in mice injected with USP7ATRT tumor cells and treated with three systemic injections of ZIKV^BR^ (Figure 5C). These transcripts were downregulated in the mock and ZIKV^BR^ systemic group, indicating an immune evasion after tumor injection that is reversed by ZIKV^BR^ infection of tumor cells leading to an inflammatory response.

In the 3D co-cultured in vitro model, we observed a clear clustering of brain organoids co-cultured with embryonal CNS tumors and submitted to ZIKV^BR^ infection (Figure 5D). There was an upregulation of IL-1b, Ltb, Tnf, and Cxcl2 in brain organoids with ZIKV^BR^ and tumor cells indicating inflammatory signaling mediated by the TNF family (Figure 5E). The presence of ZIKV^BR^ in the 3D brain organoid, with or without tumor cells, upregulated the B2M transcript, that displays anti-pathogenic activity in amniotic fluid, and downregulated the cytokine Spp1, associated tumorigenesis, and the growth factor Nodal, required for the maintenance of human embryonic stem cell pluripotency (Figure 5F). Together, B2m, Spp1, and Nodal modulation indicate a stemness and tumorigenesis impairment in both normal and tumoral models. After analyzing the healthy brain organoid independently of tumor cell co-culture, we verified that the ZIKV infection induced the overexpression of the Csf1 and the BMPs family, genes important for normal neural development, only in tumor-free organoids (Figure 5G).

## 4. Discussion

With the recent growth of viral therapies studies aiming to treat different kinds of tumors, ZIKV^BR^ appears as a promising alternative for CNS tumors, especially pediatric brain tumors with a progenitor origin. The unique ZIKV features, namely the blood-brain barrier cross capability, selective targeting of stem-like cancer cells, and the activation of an anti-tumoral immune response, turn this virus a strategic weapon against CNS tumors since the immune brain isolation and the presence of cancer stem cells make these cancers untreatable due to the usual failure of conventional therapies and novel immunotherapies [4]. The use of an oncolytic virus requires safety protocols since, differently from conventional drugs, they can replicate in biological organisms. Therefore, research in in vivo and in vitro models is important to predict the virus behavior in the human body. However, it is noteworthy that the epidemic of the ZIKV infecting thousands of individuals in Brazil in 2015–2016 demonstrated that, with exception of fetuses exposed during pregnancy, the great majority of infected people remain asymptomatic or with very mild symptoms.

The determination of maximum tolerated doses (MTD) is a crucial part of preclinical drug development to define the dose and schedule for Phase I clinical trials [17]. Unfortunately, there are many limitations described in ZIKV mouse models and difficulties to determine the MTD, especially considering brain tumor preclinical models. Adult immunocompetent mice infected with ZIKV^BR^ develop a transient viremia, but do not demonstrate signs of morbidity or mortality [18]. Other species, including cotton rats, guinea pigs, rabbits, and rhesus monkeys do not develop CNS disease and a brain injury even when infected by intracerebral inoculation [19]. Immunodeficient mice models allow the investigation of the possible neurological damage and mimics the immunological disability of cancer patients’ conditions after conventional therapy [20].

Here we showed that the immunodeficient Balb/C nude mice submitted to three serial intracerebroventricular inoculations of 2000 PFU ZIKV^BR^ induced intensive brain injury leading to mice death within 4 weeks after virus administration. Surprisingly, the same ZIKV^BR^ serial doses injected by systemic internalization presented no signals of systemic infection and brain injury. The confirmation that this mice model is susceptible to ZIKV^BR^ infection enables its use in oncolytic virus studies since it allows xenograft tumor cells to grow. Another highly susceptible to ZIKV infection animal model, the IFN-α/β receptor (IFNAR)-deficient mice are deficient in the type I or type II interferon response and develop a severe neurological disease due to ZIKV infection [21]. However, (IFNAR)-deficient mice allow only murine tumor cells in vivo growth and not human-derived tumors; therefore, they are not a good model for preclinical studies.

When the Balb/C nude mice bearing the embryonal CNS tumor cell USP7ATRT were injected with ZIKV^BR^ they showed a significantly increased survival rate and presented virus replication preferably in the brain. The histological analysis in previous studies showed that the virus naturally targets the tumor cells, specially USP7ATRT that presents neural progenitor molecular fingerprints [3]. Here, we showed that when the tumor is located subcutaneously in the animal flank, the systemically injected ZIKV^BR^ loses its tropism towards the tumor and is found in the brain. The virus preferably infects the tumor cells growing in the cerebroventricular locus inducing intensive cell death.

Besides the Balb/C nude mice immunodeficiency due to thymus absence, the cytokines and chemokines phenotype after ZIKV^BR^ systemic treatment showed the downregulation of metastasis inducers, GPI1 and VEGFA, and the macrophages migration inhibitory factor (MIF), key regulators in tumorigenesis, angiogenesis, and tumor metastasis [22]. This is important evidence that ZIKV^BR^ can block tumor immune evasion by targeting MIF, a factor that inhibits the increase in CD8 T cells number in the tumor microenvironment and reduces immune suppression in glioblastoma (GBM) [23].

The in vitro 3D brain organoid models co-cultured with embryonal CNS tumor cells also showed ZIKV^BR^ tropism to the tumor cells and an immune response mediated by TNF signaling, an important pathway that enhances the sensitivity of tumors to immunotherapy, including medulloblastoma [24]. The early-stage human 3D organoid was also a relevant in vitro model suggesting virus preference to tumor cell even co-cultured with SOX2^+^ normal cells. These are intriguing results since studies have shown that the virus preferentially infects and kills SOX2^+^ cells [2]. The organoid infection induced the downregulation of NODAL, an important growth factor for pluripotency stem cell maintenance [25], impairing appropriate neural development. In tumor-free organoids, we observed the overexpression of the BMP family after virus infection, which induces the inhibition of neural progenitor cell proliferation and promotion of neuronal differentiation indicating a neural development impairment, corresponding with the microcephaly phenotype [26]. The same regulation was not observed in co-cultured organoids confirming the virus infection selectivity to the embryonal CNS tumor cells, sparing the healthy brain cells.

In short, taken together, these results confirm the promising effectiveness and safety of the oncolytic ZIKV^BR^ in an immunodeficient mice model which is highly sensitive to virus damage, and in an early-stage 3D organoid model with a high proportion of stem-like normal cells. The anti-tumoral immune response induction and the capability to destroy CNS tumors opens new perspectives for upcoming ZIKV^BR^ oncolytic virotherapy to be applied in patients who have no disease alternative. 

## Figures and Tables

**Figure 1 viruses-13-02103-f001:**
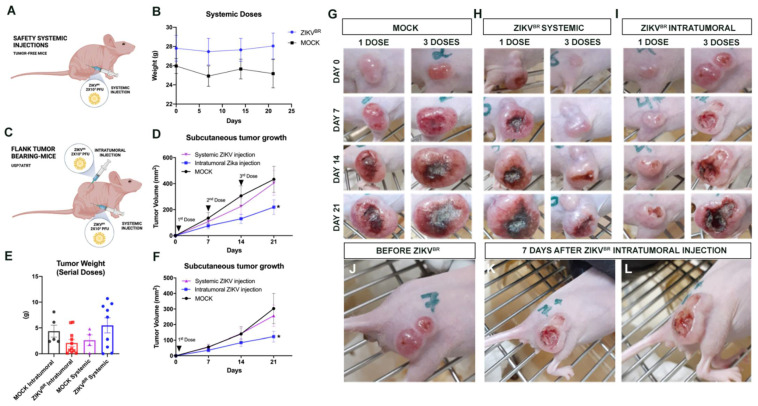
Safety serial systemic ZIKV^BR^ infections and viral tropism. (**A**) Schematic representation of tumor-free mice treated with systemic ZIKV^BR^ injections. (**B**) Weight variation in mice ZIKV^BR^ infected (2 × 10^3^ PFU/ZIKV^BR^) and control group (PBS) (*n* = 8 animals per group). (**C**) Schematic representation of flank tumor-bearing mice treated with systemic or intratumoral ZIKV^BR^ injections. (**D**) USP7ATRT tumor weight variation in the treated group, ZIKV^BR^ systemic or intratumorally injected, and its respective control groups. The tumor volume (mm^3^) and growth (doubling time (days)) in the control (mock) and ZIKV^BR^-treated groups, three doses systemic or intratumorally injected (**E**) or (**F**) a single systemic or intratumorally injection. Tumor variation after PBS injection (mock) (**G**), systemic (**H**) or intratumoral (**I**) ZIKV^BR^ single or multiple doses. Tumor establishment (**J**) and regression (**K**,**L**) observed after ZIKV^BR^ intratumoral injection. * *p* < 0.05; Student *t* test.

**Figure 2 viruses-13-02103-f002:**
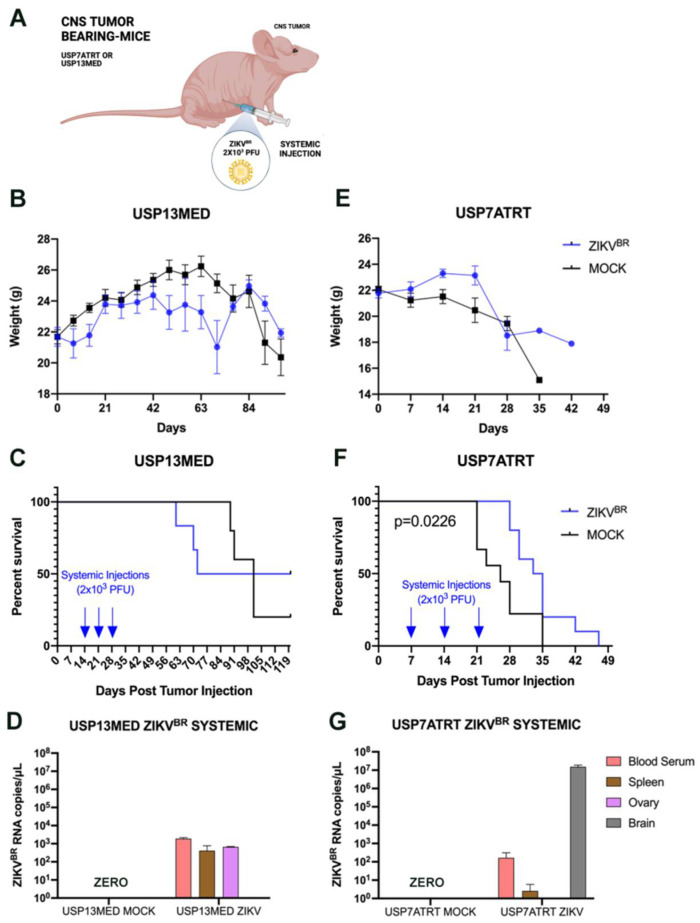
ZIKV^BR^ serial treatment through systemic injections. (**A**) Schematic representation of tumor-bearing mice treated with systemic ZIKV^BR^ injections. Bodyweight (**B**), overall survival rate (**C**), and ZIKV^BR^ RNA were detected by RT-PCR (**D**) in USP13MED tumor-bearing mice treated with three systemic ZIKV^BR^ injections. Bodyweight (**E**), overall survival rate (**F**), and ZIKV^BR^ RNA were detected by RT-PCR (**G**) in USP7ATRT tumor-bearing mice treated with three systemic ZIKV^BR^ injections. (*n* = 8 animals per group).

**Figure 3 viruses-13-02103-f003:**
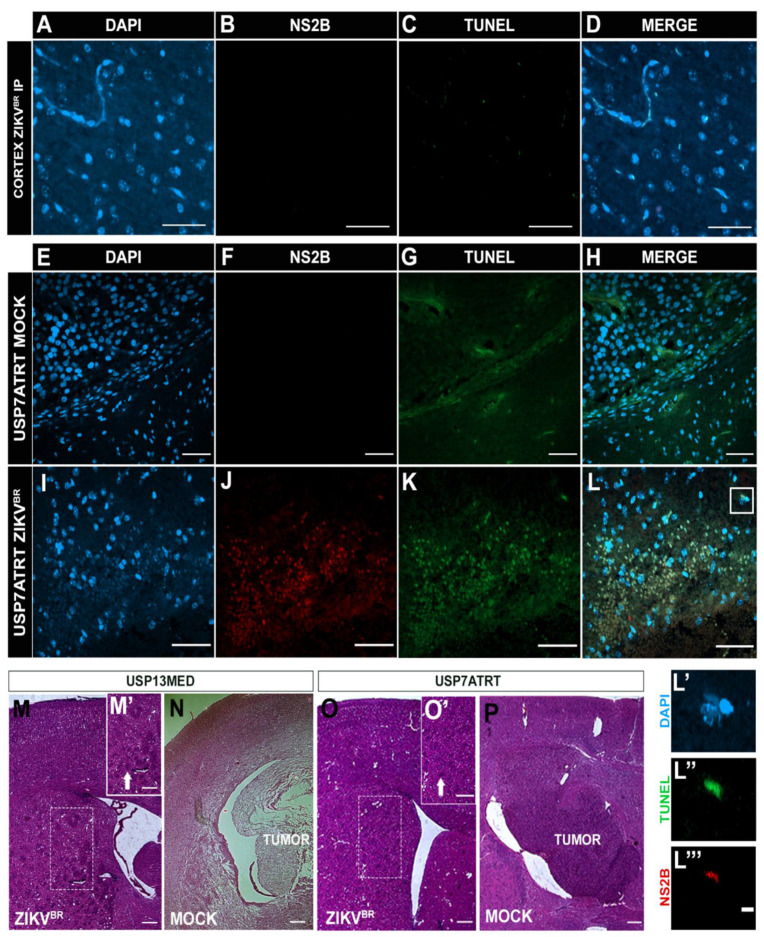
Histological analysis of mice bearing intracranial embryonal CNS tumor after ZIKV^BR^ serial treatment through systemic injections. (**A**) Immunolabeling positive of DAPI (blue), (**B**) cell death (TUNEL, green), (**C**) nuclei ZIKV^BR^ (NS2B, red), and (**D**) the merged channels (MERGE) from the cerebral cortex and (**E**–**L**) tumor edge of USP7ATRT tumor-bearing mice after systemic ZIKV^BR^/MOCK injections. Scale bar, 50 µm. Split channels co-located by DAPI (**L’**), TUNEL (**L’’**) and NS2B (**L’’’**). Scale bar, 10 µm. Histological analysis by hematoxylin and eosin stain from the brain of USP13MED tumor-bearing mice after ICV ZIKV injections (**M**), MOCK (**N**), and from USP7ATRT ICV ZIKV treated (**O**) and untreated tumor-bearing mice (**P**), highlighting the pericellular edema (**M’**,**O’**, white arrow). Scale bar, 200 µm.

**Figure 4 viruses-13-02103-f004:**
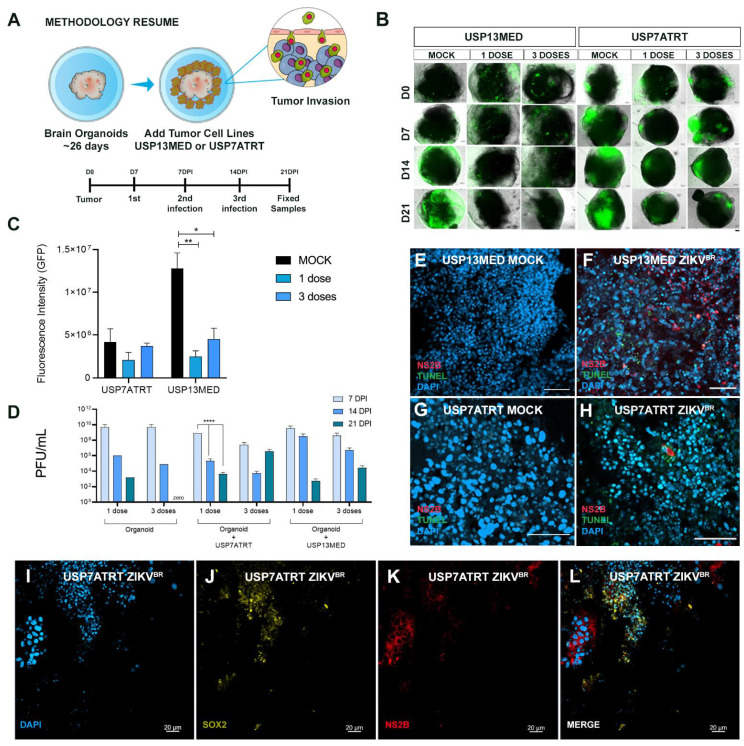
ZIKV^BR^ infection of CNS tumor cells co-cultured with brain organoids. (**A**) Schematic representation of brain organoids-CNS tumor cells co-culture assay (DPI: days post-infection). (**B**,**C**) fluorescence microscopy analysis of 26-day-old brain organoids co-cultured with 10^3^ GFP^+^ USP13MED and USP7ATRT cells and treated with none, one or three ZIKV^BR^ doses (**B**) (scale bar 20 µm). (**C**) Quantification of GFP fluorescence intensity on 21 DPI images of organoids co-cultured, *n* = 4. (**D**) Supernatant PFU assays from co-cultured treated with none, one or three ZIKV^BR^ doses at 7, 14, and 21 DPI. Immunolabeling for non-structural ZIKV^BR^ protein NS2B (red) and cell death (TUNEL, green), and SOX2 (yellow) on 21 DPI images of organoids co-cultured with USP13MED (**E**,**F**) and USP7ATRT (**G**–**L**), scale bar 50 µm (**B**,**E**–**H**) and 20 µm (**I**–**L**). (*n* = 4 per group) * *p* ≤ 0.05. ** *p* ≤ 0.01. **** *p* ≤ 0.0001.

**Figure 5 viruses-13-02103-f005:**
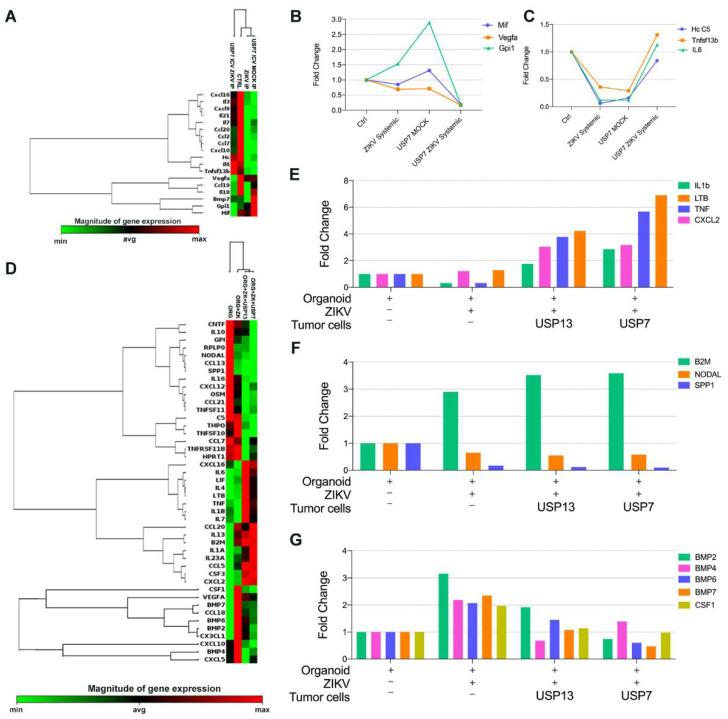
Comparative microarray gene expression of cytokine and chemokine from in vivo and 3D in vitro assay. (**A**–**C**) RT-PCR array analysis of mice brain bearing ATRT embryonal CNS tumor after three systemic ZIKV^BR^ injections (2000 PFU). (**A**) Heatmap analysis of normalized gene expression values represented as colors, from green (low expression) to red (high expression). (**B**,**C**) Highlight of transcript expression considering transcripts downregulated (**B**) and upregulated (**C**) in mice bearing USP7ATRT tumor and treated with ZIKV^BR^ in comparison with control, ZIKV^BR^, and MOCK groups. (**D**–**F**) RT-PCR array analysis of brain organoids co-cultured with/without embryonal CNS tumor cells after three 2000 PFU ZIKV^BR^ administrations. (**D**) Heatmap analysis of normalized gene expression values represented as colors, from green (low expression) to red (high expression). (**E**,**F**) Highlight of transcript expression considering transcripts upregulated (**E**) in organoids co-culture with embryonal CNS tumor cells, up/downregulated in organoids after ZIKV^BR^ infection, with and without embryonal CNS tumor cells (**F**–**G**).

## Data Availability

There is no additional data supporting reported results in this study.

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
