# Peer review of "Effect of Serial Systemic and Intratumoral Injections of Oncolytic ZIKVBR in Mice Bearing Embryonal CNS Tumors"

_viruses, 2021, doi:10.3390/v13102103_

Round 1

Reviewer 1 Report

The current manuscript describes the potential use of ZIKVBR as an oncolytic agent targeting embryonal CNS tumors. Although, the research design combining Xenograft-models with human organoid cultures appears appropriate, there are significant concerns about the benefit of ZIKVBR in the current form as an oncolytic agent. Although the virus appears capable to kill tumor cells in both the xenograft model as well as organoids, there are significant adverse effects documented and the potential benefit of the treatment remains elusive.

  • S1 clearly shows the negative impact of ZIKVBR on tumor free mice and the presence of viral RNA in Blood, Spleen and Brain. Indeed, as claimed by the authors, under such conditions, orthotopic mouse xenografts could be useful to study oncolytic ZIKV variants. However, due to the significant adverse effects the consecutive analyses performed with ZIKVBR, targeting embryonal CNS tumors are questionable, since the virus under investigation is too virulent.
  • The 3D brain model and co-culture assay, appears to be a suitable system to study oncolytic effects of any potential virus candidates, and ZIKVBR appears capable to (selectively) kill the tumor tissue and in consequence to study cytokine and chemokine expression. In addition, however, it would be more convincing to monitor the fate of healthy brain cells independent of tumor cells to exclude adverse effects on healthy brain cells.
  • It would be of interest to study less virulent ZIKV-variants with these systems as potential candidates for a treatment of embryonal CNS tumors.

Author Response

Response to Reviewer 1 Comments

The current manuscript describes the potential use of ZIKVBR as an oncolytic agent targeting embryonal CNS tumors. Although the research design combining Xenograft-models with human organoid cultures appears appropriate, there are significant concerns about the benefit of ZIKVBR in the current form as an oncolytic agent. Although the virus appears capable to kill tumor cells in both the xenograft model as well as organoids, there are significant adverse effects documented and the potential benefit of the treatment remains elusive.

Point 1: S1 clearly shows the negative impact of ZIKVBR on tumor free mice and the presence of viral RNA in Blood, Spleen and Brain. Indeed, as claimed by the authors, under such conditions, orthotopic mouse xenografts could be useful to study oncolytic ZIKV variants. However, due to the significant adverse effects the consecutive analyses performed with ZIKVBR, targeting embryonal CNS tumors are questionable, since the virus under investigation is too virulent.

Response: We agree with the reviewer that the impact of ZIKVBR on tumor-free immunosuppressed mice was too virulent. However, we need to consider Balb/C Nude unique sensibility to viral infection considering its thymus absence. There are many limitations in using murine models for ZIKV virotherapy studies. Adult immunocompetent mice infected with ZIKV develop a transient viremia, but do not demonstrate any signs of morbidity or mortality (Smith et al, 2017. doi:10.1371/journal.pntd.0005296.). In other species, including cotton rats, guinea pigs, rabbits, and rhesus monkeys, they did not develop CNS disease even when infected with ZIKV by intracerebral inoculation (Kazachinskaya et al, 2018. doi: 10.21055/0370-1069-2018-4-6-14).

The studies describe a significant virulence only in immunosuppressed models. Mice deficient in the type I or type II interferon response developed a severe neurological disease due to ZIKV infection (Smith et al, 2017. doi:10.1371/journal.pntd.0005296). In immunocompetent mice, the maximum tolerated dose (MTD) analysis was possible only after MAb-5A3 administration that blocked IFN-I signaling. The MTD observed was 9.7 days after intraperitoneal injection with 1.2 x10^6 copies of ZIKV and 14.75 days for mice exposed to subcutaneous injection.

Therefore, the problem of choosing an animal model for an oncology virotherapy study is based on the difficulty of finding an immunocompetent animal that allows tumor growth. An immunocompetent animal is needed to evaluate the immune system involvement. However, only murine cancer cells grow in the immunocompetent mice. That’s why choosing an immunocompetent mice model, that presents low ZIKV virulence will not be useful for xenografic cancer studies. Unfortunately, if we want to study human cancer cells in murine models, we need to use immunosuppressed mice which present a high virulence after ZIKV infection.

Additionally, in our previews study using dogs with spontaneous SNC tumors to test the oncolytic effect of ZIKV, no clinical sign of ZIKV systemic infection was observed, including the general symptoms that are arthralgia, myalgia, generalized weakness, vomiting, joint pain, and diarrhea, after ZIKV intrathecal injection (Kaid et al, 2020. doi: 10.1016/j.ymthe.2020.03.004). To choose the canine ZIKV dose injection, we estimated the first dose according to the animal weight: on  average weight  a dog is approximately 500 times heavier than a mouse, so the first canine dose was 1,000,000 (500 x 2,000). Thus, considering that no virulence was observed in immunocompetent animal models, the high virulence observed in the present study is a particularity of the mice models used and does not impair the potential of using ZIKV as oncolytic therapy against SNC tumors. 

Moreover, thousands of individuals were infected with zika virus during the epidemics in Brazil and it is known that the great majority remained asymptomatic or had very mild transient symptoms. Furthermore, a very recent follow-up study of children 1 to 31 months, with normal head circumference, who were infected by zika virus during pregnancy are showing normal cognitive and motor development (unpublished data).

Point 2: The 3D brain model and co-culture assay, appears to be a suitable system to study oncolytic effects of any potential virus candidates, and ZIKVBR appears capable to (selectively) kill the tumor tissue and in consequence to study cytokine and chemokine expression. In addition, however, it would be more convincing to monitor the fate of healthy brain cells independent of tumor cells to exclude adverse effects on healthy brain cells.

Response: We thank the reviewer for the suggestion. We agree with the reviewer that we could better explore these results. In fact, after performing independent analyses in the healthy brain organoid we observed that the ZIKV infection induced the overexpression of genes involved with neural development. That’s why we included a new graphic (Fig 5-G) highlighting the overexpression of cytokines observed only in the tumor-free organoid infected with ZIKV. As suggested, we also discussed the findings in section 4:

“This early-stage human 3D organoid was also a relevant in vitro model for non-tumoral neural cells ZIKVBR infection. In tumor-free organoids, we observed the overexpression of BMP family after virus infection, which induces inhibition of neural progenitor cell proliferation and promotion of neuronal differentiation indicating a neural development impairment, corresponding with microcephaly phenotype (Gámez et al, 2013. doi: 10.1016/j.ymthe.2020.03.004).  The same regulation was not observed in co-cultured organoids confirming the virus infection selectively to the embryonal CNS tumor cells, sparing the healthy brain cells.”     

Point 3: It would be of interest to study less virulent ZIKV-variants with these systems as potential candidates for a treatment of embryonal CNS tumors.

Response: We appreciate the reviewer’s suggestion. We believe that, based on the literature reports, we are working with the less virulent ZIKV-variant. The infection of the immunosuppressed mice model with subcutaneous injection of African and Asian ZIKV strain induces significant weight loss and animal death after 10 and 6 days, respectively (Simonin et al, 2017. doi:  10.1371/journal.pntd.0005821.). In our study, we observed no side effects after Brazilian subcutaneous injection in Balb/C Nude mice. Since we aim to use the virus as oncolýtic therapy against CNS tumors, it is important to work with a low virulent ZIKA strain.

Reviewer 2 Report

The manuscript entitled “Effect and safety of serial systemic and intratumoral injections of oncolytic ZIKVBR in mice bearing embryonal CNS tumors” by Ferreira et al. covers the effect and safety of serial systemic and intratumoral injection of oncolytic ZIKVBR using embryonal CNS tumors-bearing mice. The manuscript is interesting to see the potential use of ZIKV for the treatment of embryonal CNS tumors, but there are some issues to be considered for publication, as follows:

  1. Abstract: authors are recommended to revise the abstract more impact. Please make clearer in terms of the rationale of this study and how this manuscript solves the issues and why it is meaningful(mechanism) for the treatment. Please check the sentence: “The Zika virus (ZIKV) has shown a promising oncolytic effect against embryonal CNS tumors. However, studies on the effect of different administration routes and the ideal viral load in 12 preclinical models are highly relevant.” Do you mean that the effects are different according to the administration routes and viral load? If so, related result and conclusion of why and how should be clearly presented.
  2. In materials and methods, the information of Zika virus used in this study and cell lines is limited. Although authors quoted the reference of the source, please don’t miss import information and why they should be used instead of commercial cell lines. “The human induced pluripotent stem (hiPS) cell line 65 used in this experiment was previously established and characterized by our group [6] (line 65)”: Which cell lines were used? hiPS cell line establishment can be introduced shortly instead of just quoting ref. What is i9048 cell line?(line 68). Please introduce briefly about USP7-ATRT and USP13-MED. Is there no comparable commercial cell lines? And I am not sure both cells are tumor and how they are established and what is different (nomal vs tumor? Or medulla vs different origin but established in brain?) Some materials used in this study need cat. numbers. 000 hiPS cells (in line 84): is it 9X103…?
  3. In results, It is not clear and difficultly readable. How about drawing the study protocol for the safe and effect study according to injection method, viral load (local vs systemic, one dose vs serial dose) and how they are analyzed and compared. (Figure 1 and 1S)
  4. To check the viral tropism, why did authors choose intracerebroventricular (ICV) injections? And why they show different result (no weight loss and no viral detection in systemic injection, but not in ICV injection.) Authors chose ip as systemic injection? Intratumoral is local injection? All are not clear to me. How they can be comparable? If authors couldn’t find viral detection in system injection, authors need to consider if it is because of injection method (maybe they need to iv injection) or viral doses (2X103 pfu should be compared with several higher doses. Eg. 2000, 20000, 200000 pfu). Some refs say that ZIKV has tropism to neural tissue, ocular tissues, testis…. Author should check other references and chose right model of injection and viral doses and comparative group. And there is no information regarding how many mice per group are used.
  5. Author said viral tropism for the mice CNS, but why no virus was detected in systemic injection (Fig. 1)? Does ZIKABR has CNS tropism or Tumor tropism? It is still not clear.
  6. Intracranial injection (serial) caused toxicity to tumor free mice (Figure S1), which means that the viral doses used are toxic to normal mice? (dead in 28 days after injection), which means that ZIKA is more toxic to normal mice than brain tumor is. (Figure 2, compare to no viral injection group. They survived longer).
  7. Authors can add fluoresce intensity graph to confirm their expression and localization in Figure 3. Authors are recommended to arrange images to be easily compared (It is better that H&E and immunostaing image with the same scale and location should be shown to be compared).
  8. In Figure 4, why there is no different between 1 dose and 3 doses? Is it same with animal model? It seems that Organoid are affected by viral treatment (Figure 4D). Indication of statistically difference seems not matched with the graphs shown.
  9. From results, it is still not clear that viral tropism (CNS? Or Tumor?). Neither is regarding which viral load, doses and injection (systemic vs local) method used.
  10. With reasonable data display together with expression profile, authors are recommended how viral load and doses will work for CNS tumors.

Author Response

Response to Reviewer 2 Comments

The manuscript entitled “Effect and safety of serial systemic and intratumoral injections of oncolytic ZIKVBR in mice bearing embryonal CNS tumors” by Ferreira et al. covers the effect and safety of serial systemic and intratumoral injection of oncolytic ZIKVBR using embryonal CNS tumors-bearing mice. The manuscript is interesting to see the potential use of ZIKV for the treatment of embryonal CNS tumors, but there are some issues to be considered for publication, as follows:

Point 1: Abstract: authors are recommended to revise the abstract more impact. Please make clearer in terms of the rationale of this study and how this manuscript solves the issues and why it is meaningful (mechanism) for the treatment. Please check the sentence: “The Zika virus (ZIKV) has shown a promising oncolytic effect against embryonal CNS tumors. However, studies on the effect of different administration routes and the ideal viral load in preclinical models are highly relevant.” Do you mean that the effects are different according to the administration routes and viral load? If so, related results and conclusion of why and how should be clearly presented.

Response: We appreciate the reviewer’s suggestion. The abstract was revised in the updated manuscript file.

Pont 2: In materials and methods, the information of Zika virus used in this study and cell lines is limited. Although authors quoted the reference of the source, please don’t miss import information and why they should be used instead of commercial cell lines. “The human induced pluripotent stem (hiPS) cell line used in this experiment was previously established and characterized by our group [6] (line 65)”: Which cell lines were used? hiPS cell line establishment can be introduced shortly instead of just quoting ref. What is i9048 cell line?(line 68). Please introduce briefly about USP7-ATRT and USP13-MED. Is there no comparable commercial cell lines? And I am not sure both cells are tumor and how they are established and what is different (nomal vs tumor? Or medulla vs different origin but established in brain?) Some materials used in this study need cat. numbers. 000 hiPS cells (in line 84): is it 9X103…?

Response: As described in the original manuscript, the Zika virus used in the study was donated by Dr. Pedro Vasconcelos (Evandro Chagas Institute, Brazil). We included the sequence GenBank accession number ID (KU497555) in the updated manuscript to give more details about the Brazilian virus strain used in the present study.

We described better the hiPS origin in the updated manuscript as required by the reviewer: hiPS cell line used in this study was generated by reprogramming of skin fibroblasts from a 26 years old healthy men. F9048 cell line presents pluripotent cell colony morphology and positive staining for markers OCT3/4 and SSEA-4 (Miller et al,. 2017. doi:10.1093/hmg/ddx078 )”.

Regarding the embryonal CNS tumor cells, all the information required by the reviewer were already present in the original manuscript in the Materials and Methods section: “The embryonal CNS tumor cell lines USP7-ATRT (atypical teratoid/rhabdoid tumor, in-house established; [Kaid et al, 2018. doi:10.1158/0008-5472.CAN-17-3201], USP13-MED (medulloblastoma, in-house established); [Silva et a, 2015. doi: https://doi.org/10.1007/S10616-015-9914-5]”. More information about the in-housing tumor cell lines can be found in the original papers (ref 3 and 7, also cited here) that describe in detail the cell line establishment and characterization.

Point 3: In results, It is not clear and difficultly readable. How about drawing the study protocol for the safe and effect study according to injection method, viral load (local vs systemic, one dose vs serial dose) and how they are analyzed and compared. (Figure 1 and 1S).

Response: We thank the reviewer for the suggestion. We included a schematic figure for all methods in the updated manuscript to facilitate the study comprehension.

Point 4: (i) To check the viral tropism, why did authors choose intracerebroventricular (ICV) injections? (ii) And why they show different result (no weight loss and no viral detection in systemic injection, but not in ICV injection).

Response: The first aim of the present study was to analyze the ZIKV tropism. For that, we challenge the virus, systemically internalized, in a mouse model with subcutaneous embryonal CNS tumor cells to check if the virus would present a tropism for the CNS or for cancer stem cells growing outside of the CNS. Additionally, we included a positive control group submitted to ZIKV intratumoral injection to observe the oncolytic effect of ZIKV directly injected into the tumor mass. As shown in Figure 1, we confirmed that the viral tropism was directed  to the brain. However, the intratumoral injection showed significant tumor destruction. That’s why we decided to investigate the intracerebroventricular (ICV) ZIKV treatment since this virus internalization method better mimics an “intratumoral” injection for embryonal SNC tumors.

The second aim was to evaluate the safety and effectiveness of intratumoral/intracerebroventricular ZIKV injection in mice bearing CNS tumors. For that, we used the metastatic orthotopic xenograft mouse model, first in a tumor-free mice group (Figure S1) and then in mice bearing CNS tumor group (Figure S2). Unfortunately, the ICV internalization method showed high virulence.

Since the systemic virus internalization method showed no side effects correlated with ZIKV infection (Fig 1A), we followed the third aim: to evaluate effectiveness of a systemic ZIKV treatment in mice bearing CNS tumor. Here, we used the same metastatic orthotopic xenograft mouse model of aim 2, however with another virus internalization method. In Figure 2, we show that the systemic injection increased the mice’s survival and had no side effects.              

To avoid any misunderstanding, we explained better each section describing the aims in the updated manuscript.

(iii) Authors chose ip as systemic injection?

Response: We agree with the reviewer that IP abbreviation is confusing in the original text. We choose IP (intraperitoneal) as systemic injection. We now changed the term “intraperitoneal” for “systemic” throughout the text and in the figures. We also created a topic (2.4 in the Materials and Methods section) to clarify any doubts.

(iv)  Intratumoral is local injection? All are not clear to me.

Response: Yes, “intratumoral'' is the injection into the tumor. Here, systemic injection is used meaning “intraperitoneal injection” (IP). Intratumoral and intracerebroventricular injections are the same as local injection, but in different places. We believe that we solved this problem creating a topic to clarify this question as described above.

(v) How can they be comparable?

Response: As explained above, both systemic (via intraperitoneal) and intratumoral injections, shown in Figure 1, are comparable because the mice groups present the same subcutaneous tumor growing the flank, changing the virus internalization method.

(vi) If authors couldn’t find viral detection in system injection, authors need to consider if it is because of injection method (maybe they need to iv injection) or viral doses (2X103 pfu should be compared with several higher doses. Eg. 2000, 20000, 200000 pfu). Some refs say that ZIKV has tropism to neural tissue, ocular tissues, testis…. (vii) Author should check other references and choose the right model of injection and viral doses and comparative group.

Response: We agree with the reviewer that different injection methods and higher doses could increase the chance of the virus to find a susceptible cell to replicate its genome, allowing viral organ detection. However, since we detected the viral RNA in the brain of mice bearing intracranial tumors and also observed higher survival rate, it confirms that the systemically injection of 2x103PFU is enough to the treatment efficiency.

Considering the murine models that could be used in this study, there are many limitations in using animal models for ZIKV virotherapy studies. Adult immunocompetent mice infected with ZIKV develop a transient viremia, but do not demonstrate any signs of morbidity or mortality (Smith et al, 2017. doi:10.1371/journal.pntd.0005296). In other species, including cotton rats, guinea pigs, rabbits, and rhesus monkeys, they did not develop CNS disease even when infected with ZIKV by intracerebral inoculation (Kazachinskaya et al, 2018. doi: 10.21055/0370-1069-2018-4-6-14). The studies describe a significant virulence only in immunosuppressed models. Mice deficient in the type I or type II interferon response developed a severe neurological disease due to ZIKV infection (Smith et al, 2017. doi:10.1371/journal.pntd.0005296). Therefore, the problem of choosing an animal model for an oncology virotherapy study is based on the difficulty of finding an immunocompetent animal that allows tumor growth. That’s why choosing an immunocompetent mice model, that presents low ZIKV virulence, will not be useful for xenographic cancer studies. If we want to study human cancer cells in murine models, we need to use immunosuppressed mice which present a high virulence after ZIKV infection.

Additionally, in our previous study using dogs with spontaneous SNC tumors to test the oncolytic effect of ZIKV, no clinical sign of ZIKV systemic infection was observed, including the general symptoms that are arthralgia, myalgia, generalized weakness, vomiting, joint pain, and diarrhea, after ZIKV intrathecal injection (Kaid et al, 2020. doi: 10.1016/j.ymthe.2020.03.004). Thus, considering that no virulence was observed in immunocompetent animal models, the high virulence observed in the present study is a particularity of the mice models used and does not impair the potential of using ZIKV as oncolytic therapy against SNC tumors.  

(viii) And there is no information regarding how many mice per group are used.

Response: We thank the reviewer for the observation. The total and the number of mice used for each assay were included in Materials and Methods (topic 2.4) and in all figure legends in the updated manuscript.

Point 5: Author said viral tropism for the mice CNS, but why no virus was detected in systemic injection (Fig. 1)? Does ZIKABR has CNS tropism or Tumor tropism? It is still not clear.

Response: The ZIKV tropism to CNS is well established in literature. We better answer this question at the “Point 9”, below. Our results cannot confirm that there was no viral replication. What they show is that 7 days after the infection no viral particles were found. Replication may have occurred, but at such low rates that it was not detected. However, it was possible to detect the viral particles when in the presence of tumor cells - as, we believe, replication rates (confirmed by immunohistochemical images) are high enough to be detected by RT-PCR.

Point 6: Intracranial injection (serial) caused toxicity to tumor free mice (Figure S1), which means that the viral doses used are toxic to normal mice? (dead in 28 days after injection), which means that ZIKA is more toxic to normal mice than brain tumor is. (Figure 2, compare to no viral injection group. They survived longer).

Response: We agree with the reviewer that the ZIKV toxicity in tumor-free mice is higher than in mice bearing CNS tumor cells is intriguing. That's why we better investigated this matter in our previous study published by Cancer Research in 2018 (DOI: 10.1158/0008-5472.CAN-17-3201). Using the same embryonal CNS tumor cell lines, we observed that the ZIKV derived from tumor cell infection was not capable to infect Vero cells and neither normal neuron, presenting a lower PFU titer. We believe that in tumor-free mice the virus can infect the normal CNS cells leading to the ZIKV high virulence. However, in the presence of a tumor, the virus selectively infects the tumor cells, as shown in the Figure 3, and releases to the microenvironment “defective” virus incapable of reinfecting the normal CNS cell, avoiding ZIKV toxicity.

Point 7: Authors can add fluorescence intensity graph to confirm their expression and localization in Figure 3. Authors are recommended to arrange images to be easily compared (It is better that H&E and immunostaing image with the same scale and location should be shown to be compared).

Response: We thank the reviewer’s suggestion. Unfortunately, the deadline of 10 days to revise the manuscript given by the journal is not enough time to perform this analysis. However, we believe that the images already included in the study present enough data to support the manuscript conclusions.

Point 8: In Figure 4, why there is no different between 1 dose and 3 doses? Is it same with animal model? It seems that Organoid are affected by viral treatment (Figure 4D). Indication of statistically differences seems not matched with the graphs shown.

Response: As mentioned in section 3.3, the absence of difference between 1 and 3 doses are due to the high virus titer present in the supernatant before addition of the second and third dose in the culture: “The additional 2,000 PFU doses seems to add no significant effect since the culture supernatant viral titer is already extremely high (Figure 4D).”

Point 9: From results, it is still not clear that viral tropism (CNS? Or Tumor?). Neither is regarding which viral load, doses and injection (systemic vs local) method used.

Response: Taking together all the results, we demonstrated in the present study that when the virus is injected systemically in mice bearing a subcutaneous tumor, the tropism is towards the SNC (Figure 1). When the tumor is growing in the cerebral ventriculum, the virus tropism is towards cancer cells, sparing the normal brain cells (Figure 3). In the 3D co-culture organoid model, we observed that even in the presence of progenitor health brain cells, the virus selectively infects the tumor cells (Figure 4 and 5).  So, the results reinforce the potential of ZIKV as oncolytic therapy against SNC tumors, confirming the systemically injection and multiple doses as viable clinical approaches.

Point 10: With reasonable data display together with expression profile, authors are recommended how viral load and doses will work for CNS tumors.

Response: We agree with the reviewer and thank for the suggestion. We intend to continue the study to better understand the ZIKV mechanism according to different viral loads and doses.

Round 2

Reviewer 1 Report

The efforts explaining the drawbacks of preclinical assessments are indeed appreciated and it is indeed a challenge to choose preclinical models to determine efficacy and safety of a cancer therapy approach with human viruses. However, the mouse model used for this study does not demonstrate safety of ZIKVBR (as stated in the title), and it still appears very difficult to assess, whether normal human astrocytes and other cerebral cells within the organoids remain unaffected by the virus treatment.

Author Response

Reviewer 1

The efforts explaining the drawbacks of preclinical assessments are indeed appreciated and it is indeed a challenge to choose preclinical models to determine efficacy and safety of a cancer therapy approach with human viruses. However, the mouse model used for this study does not demonstrate safety of ZIKVBR (as stated in the title), and it still appears very difficult to assess, whether normal human astrocytes and other cerebral cells within the organoids remain unaffected by the virus treatment.

Response: We thank you very much for the distinguished reviewer, which allowed us to significantly increase the quality of the paper.  We agree with the reviewer about the lack of safety analysis regarding the normal cells health assessment. These important questions will be directed to another study to be submitted soon. 

So, as asked by the reviewer, we changed the title for “Effect of serial systemic and intratumoral injections of oncolytic ZIKVBR in mice bearing embryonal CNS tumors.”

Reviewer 2 Report

Thank authors for their efforts to revise the manuscript. 

I am just asking authors to reflect their responses also on the revised version again, not just merely answering to the comments. In addition, please let me know how they reflected in the mansucript by quoting "...... (line # in page #)" in the responding letter.

Author Response

Reviewer 2

Thank authors for their efforts to revise the manuscript. 

I am just asking authors to reflect their responses also on the revised version again, not just merely answering to the comments. In addition, please let me know how they reflected in the mansucript by quoting "...... (line # in page #)" in the responding letter.

Response: 

We appreciate the reviewer’s suggestions. 

In the last manuscript version, we performed these changes:

  • The abstract has been extensively modified as suggested, to highlight the aims of the study;
  • Section 2.1: we have added detailed information on F9048 cells description in materials and methods;
  • Methodological representation has been added to each figure for a better understanding of the experimental design in all figures;
  • Section 2.4: we changed “IP” to “Systemic Injection” to clarify the experimental design understanding. The chance was performed on all figures and graphics;
  • In the first paragraph of all the results sections, a sentence was included describing the aims of the proposed assays;
  • Section 3.4: we included a new graphic (Figure 5G) describing the infection effect in the healthy brain organoid. The discussion of this result was included in the last but one paragraph of section 4.